# Development of Ontology for Knowledge of Traditions Common Culture of Countries in the Greater Mekong Subregion

**Suwannee Hoaihongthong** and **Kanyarat Kwiecien** *

Department of Information Science, Faculty of Humanities and Social Sciences, Khon Kaen University, Khon Kaen 40002, Thailand
* Correspondence: kandad@kku.ac.th; Tel.: +66-84-600-9643

**Abstract:** The development of ontology is one important research area in the digital humanities. This study aims at creating a semantic search system for traditions common culture in the Greater Mekong Subregion (GMS) to solve problems in semantic gaps. This paper presents the second phase of the main research. It will present how to develop ontologies for the traditions and common culture in the GMS to gain a perspicuous understanding of the traditions and common culture of those countries in the region. A theoretical concept of seven steps for ontology development was applied by using an ontology editor called Hozo Ontology Editor. The main ontology found in this study included 15 main classes: common culture, history, belief, purpose, location, ritual, activity, literature, values, place, time, principle, person, equipment, and ethnic group. Traditions common culture is a subclass of common culture classes that were found to be related to all classes in ontology. This ontology will be useful for developing a semantic search system of the traditions common culture of the GMS in the next steps of the main study.

**Keywords:** ontology; common culture; traditions common culture; knowledge organization

## 1. Introduction

Countries in the Greater Mekong Subregion (GMS) include the Kingdom of Cambodia, the People's Republic of China, the Lao People's Democratic Republic, the Republic of the Union of Myanmar, the Kingdom of Thailand, and the Socialist Republic of Vietnam. These countries have had international relations with one another for a long period of time with respect to the economy, society, politics, and culture. Moreover, they have continually expanded cooperation in various areas, especially, the Greater Mekong Subregion economic cooperation program implemented since 1992 [1,2].

An underlying strong point of the GMS countries is cultural affinity or commonalities, also referred to as common culture. In fact, even if these countries have evolved in a variety of aspects according to history, politics, and governance, their identity, selfness, and ethnicity still remain unaltered [3]. These so-called cultural commonalities affect the people's way of life, local wisdom, and expression of arts and culture, which lay the foundation for a regional identity [4]. Culture not only portrays the prosperity of a society in each locality, but also serves as a vital foundation of the entire culture. In the world's cultural trends, which are being globalized, the expression of "affinity" and the portrayal of local cultural diversity are "differences" that are vital for adaptation, a charm, and the true self of each society and culture. Hence, the realization of values of local cultures with diversity can contribute to sustainable social-cultural development in every aspect, in compliance with the conditions of the era.

Common culture can be defined as the shared cultural foundation or cultural similarities of which cultural ownership is assigned to more than one party or which exist beyond ownership [5]. This form of culture refers to cultural features shared among each country

such as history, values and principles, purposes and sense of mission, and symbols and boundaries [6]. Among the GMS countries with cultural commonalities are Thailand, the Lao People's Democratic Republic, and the Kingdom of Cambodia. Simply put, they share cultural roots due to geographical factors, cultural foundations, religion, and similar ways of life. In addition, they transmit culture to one another through visits, trade, and relocation, which bring about similar practices and the same trends. Common culture is regarded as an important component in public diplomacy policies and is employed to strengthen international relations in order to maintain each country's benefits. As well-documented, culture is used to establish international relations to carry out tourism policies and to develop strategies for the promotion of tourism, especially cultural tourism, in order to generate incomes for these countries [7]. Apart from this, common culture is associated with economic policies, that is, attention is given to the promotion of the creative, culture, and cultural services industry (e.g., food, films, fashion, fighting, and festivals as well as cultural products of Thailand). Most importantly, the ASEAN Socio-Cultural Community Blueprint has been created in order to create a community with interaction, awareness, and pride in its identity and culture [4].

Despite the various cooperation policies and cultural cooperation framework, no international organizations have been clearly assigned to conduct research, store knowledge, and organize knowledge about the common culture among the GMS countries. Access to cultural knowledge falls under the responsibility of each country with its own organization assigned to collect the cultural information of particular countries [8]. For example, the access to cultural information in Laos is similar to that in Cambodia. That is, the information is stored at the Ministry of Culture and Fines Arts, and website of the Heritage and Arts Information Center provides information services; nevertheless, its presentation lacks knowledge organization, links, and search tools [9].

The results of previous studies on the search systems of knowledge about common culture among the GMS have shown that general search systems have been found to be rather restrictive. The system searches for data or documents have mainly used character comparison methods. The system did not consider the exact concept or meaning of the search query. Some search terms contained inappropriate attributes or could not represent the content, mainly due to a huge gap between what the computer can interpret and what humans understand, known as the semantic gap [10]. The semantic gap characterizes the difference between two descriptions of an object by different linguistic representations (e.g., searching for traditions by name). The tradition may have different names according to languages and dialects such as Boon Pha Wet, Bun Pha Wet, Bun Phawet, Boun Phavet, The Traditional Mahajati Preaching, Mahachat, etc. Thus, a semantic search system could be a solution to this problem because it uses a string matching method and also considers synonyms and related terms. For example, to search for traditions based on the attributes of culture, a search for traditions from the Naga belief can be used. The search results will show the traditions associated with that belief. Therefore, this present study applied ontology concepts to design a structural model of knowledge regarding the traditions in GMS. This ontology will be used in the further development of semantic search systems.

Knowledge organization involves determining the concepts and semantic relations of a particular issue and establishing their relations with relevant concepts. It serves as a foundation of knowledge structure development and ontology development, which contributes to effective information search and access to knowledge [11]. Apart from studies for cultural knowledge organization, there exists an approach to cultural information management, referred to as digital humanities research. It integrates information technology into humanities for the storage, dissemination, and search of information [12]. The scope of digital humanities research derived from the integration of knowledge in the field of information science covers organization, retrieval and access, and services; moreover, its research and development approaches concern knowledge organization, ontology development, and semantic search development [13].

A knowledge organization system is the key element of knowledge engineering. Ontology provides a fundamental framework for the development of the semantic web. A knowledge organization system is the key element of knowledge engineering. Ontology provides a fundamental framework for the development of the semantic web. Knowledge organization, ontology development, and the development of semantic search are interrelated approaches. That is, the development of semantic web or semantic search entails utilizing technology to manage data, enable computers to understand information, and allow for access via the Internet; this can be achieved by data structure management to establish relationships of data using ontology and sharing data at a metadata level. Users can access this semantic search via websites. The semantic search allows people to access and make use of digital cultural content effectively [14]. In the development of a semantic search, its basic data are derived from the data structure design with ontology. Still, ontology development requires analyzing and synthesizing data and knowledge about a particular topic as well as determining the concepts and relations of knowledge systematically [15].

An ontology is a commonly used tool to explain the representation of knowledge. The advantages of an ontology include its interoperability to share a common understanding among people or software agents, it enables the reuse of domain knowledge, and facilitates explicit conventions. The digital cultural heritage ontologies are one of the most challenging methods in information management. The digital cultural heritage field has gained wide interest, both by the general public and researchers, and has a lot of heterogeneity in terms of content and its potential uses [16]. We have adapted some of the data from previous ontology to fit the scope of this study. Ontologies can be shared between applications or reused because ontology is the keyword source used to describe domain concepts. There is an index storage format that can be shared between applications. The ontology can copy the knowledge base and use it in new applications. There may be an improvement in the structure or the addition of more knowledge [17].

In this study, we examined previous existing ontologies to present a relevant academic discussion on traditions common cultural information in the context of the GMS. The purpose of developing an ontology for cultural heritage is its integration ontology into semantic applications. The application provides easy, quick, and intelligent access to the construction of project information through a user-customized definition search. Previous studies on cultural heritage ontologies have developed ontologies for the representation of Twelve-month Isan Merit-Making Traditions [18,19]. However, its research margin was narrower than the current research with regard to the context and the scope of the study. Another study is The Event Ontology, developed at the Center for Digital Music in Queen Mary, University of London, which developed a vocabulary to describe events. This ontology is centered around the notion of event, seen here as the way by which cognitive agents classify time/space reactions. It is certain that a tradition is a form of event. Therefore, this can be applied to the ontology to describe traditions in terms of time and place [20]. The DBpedia Ontology is generated from the created specifications in the DBpedia Mappings Wiki. We applied the properties of Location Ontology including locationCity and locationCountry to describe the location of the tradition [21]. Furthermore, a study by Tuamsuk, Chansanam, and Kaewboonma on cultural heritage ontologies focused on the ontologies of folktales in the GMS. The folktale domain, as a part of the literature, is vast in scope and divergent in terms of concepts and conceptual relations and is also related to traditions [22]. Finally, we also applied the ontology of cultures and ethnic groups by theory type with record types implemented by functional programming [23] and a study by Chaikhambung and Tuamsuk on ethnic ontology is an ontology that describes an overview of one's ethnic group such as the religion, belief, habitat, dress, social order, art, and history of an ethnic group. The scope of this research is different from that research. Therefore, this research applied the ontology from the ethnic ontology only for the properties of the ethnic group classes including ethnonym, exonym, and autonym [24]. Obviously, we placed an emphasis on the approach of considering reuse ontology to connect the GMS traditions

common culture ontology with previous ontologies. The disparity between this research and the previous research lie in the differences in the research focus. This study examined the traditions common culture of the GMS, which is wider in scope than the previous one. In addition, the content used in the present study is more explicit on traditions and its details can be described as the coherence of traditions in the GMS in terms of the history, values and principles, purposes and sense of mission, and symbols and boundaries. All the above information is useful for searching. More significantly, an ontology of the knowledge of traditions common culture in the GMS has yet to be created.

In this study, knowledge organization, ontology development, and the development of semantic search focused on traditions common culture. It can be extrapolated that traditions are crucial and available in all nations and languages; despite differences depending on each locality, their existence is necessary. Apart from that, they are accepted among people in society, practiced, and inherited from generation to generation. In fact, the traditions of each nation portray the concepts and beliefs through their lifestyles, which are part of its civilization and social heritage, comprising the knowledge and expertise of people as a member of society [25]. Additionally, they serve to unite members of the society in order to promote bonds and harmony as well as to cultivate shared attitudes, beliefs, and values. Thus, traditions can clearly reflect cultural commonalities. However, the study was limited to Thailand, the Lao People's Democratic Republic, and the Kingdom of Cambodia. The reason was that these countries had markedly cultural commonalities, mountain-plain border areas, the Mekong River that connects their culture, and a shared border, so citizens of these countries clearly had lifestyles, culture, and traditions in common.

The study was based on Hjørland's concept of knowledge organization [11]. Specifically, categories of knowledge were identified by determining the concepts and semantic relations as well as establishing their relations with relevant concepts. The results were presented in the form of classification schemes by displaying the data of main concepts and roughly sorting a data hierarchy according to the groups of related contents. The structure of the data was later developed into ontology. The domain ontology development was grounded on Uschold and King's concept [26] with three processes: determining the purpose and scope; ontology development; and ontology evaluation. In particular, Noy and McGuinness's seven steps for ontology development [15] was applied, and the ontology editor used in this study was Hozo Ontology Editor. Hozo is a Japanese graphical ontology editor specifically created to produce heavy-weight ontologies and was developed through a partnership between the Department of Knowledge Systems (Mizoguchi Laboratory), ISIR-Osaka University, and Enegate Co Ltd. (Osaka, Japan) [27], which has been created and evaluated by domain experts and application-based evaluation methods.

The goal of this research was to apply digital humanities research concepts in developing ontologies for the traditions common culture of the GMS. The ontologies will focus on intangible cultural heritage (ICH) and are intended to be used as information resources and information retrieval. This paper presents the second phase of the main research, focusing on the development of ontologies for traditions common culture in the GMS. As a knowledge organization system, ontologies can be regarded as the classification of knowledge that provides the scope, concepts, and structure of the traditions common culture in the GMS, which in turn, will be useful for understanding and searching the traditions common culture in the GMS information. Concepts of knowledge organization and ontology development can link related knowledge systematically and allow for the sharing of data, while a semantic search will allow users to access and make use of data [28]. The developed ontology will be useful for the development of a semantic search system of the traditions common culture of the GMS to solve the problem of semantic gaps in the next steps of this research.

## 2. Research Objectives and Methods

The present study sought to develop ontology for knowledge about the traditions common culture among the countries in the GMS in order to describe the scope of the

knowledge and to create a clear understanding of the traditions common culture of those countries. The ontology in this study will be used to further develop semantic search systems.

Uschold and King's concept of domain ontology development [26] was adopted as a framework for ontology development, which consisted of the following processes: (1) determination of purposes and scope; (2) ontology development; and (3) ontology evaluation. The current study presents the main findings from the second process of the study, the ontology development process. The research conceptual framework is presented in Figure 1.

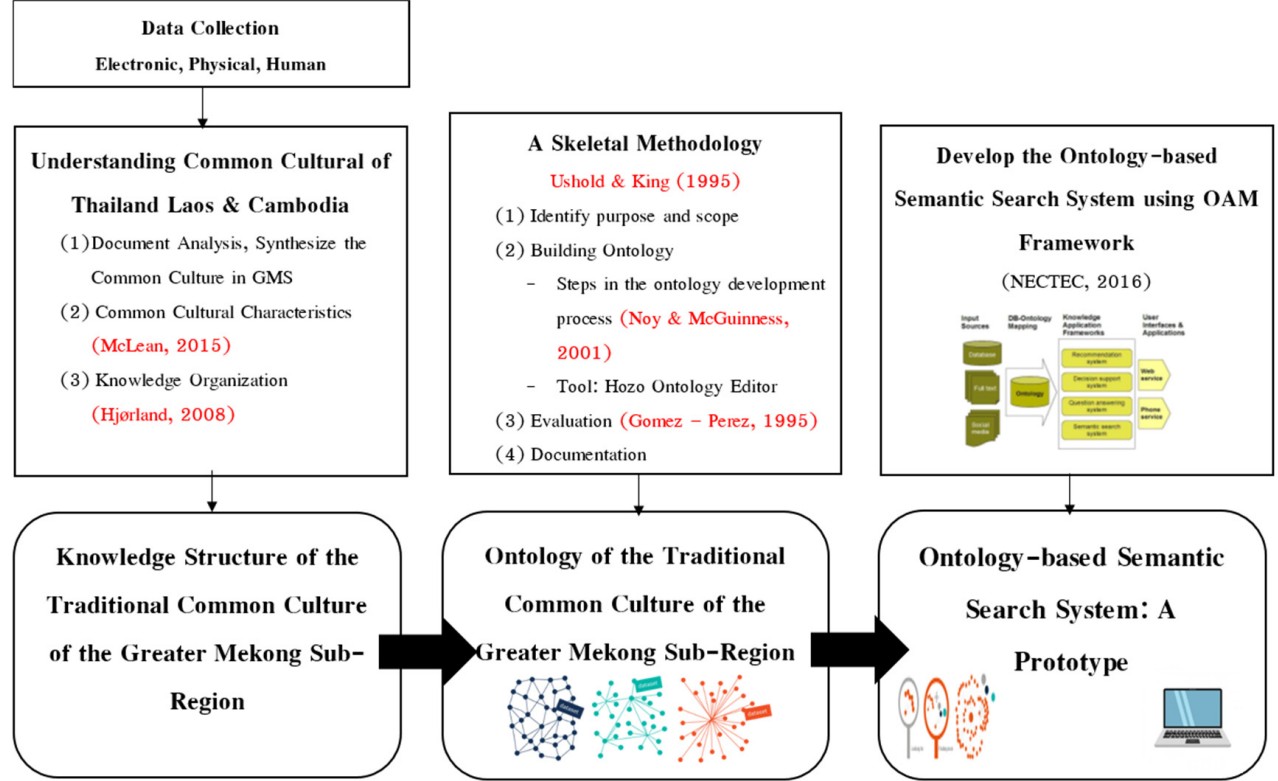

**Figure 1.** The research conceptual framework.

### 2.1. Determination of Purposes and Scope

This ontology development study aimed to describe the scope of knowledge and to create a clear understanding of the traditions common culture of the GMS countries. Three of the countries that were the main focus of this study included Thailand, Laos, and Cambodia. These three countries were selected because they share land and river borders including mountains, plains, and the Mekong River, which is an important river in the region. Therefore, common ways of the life, cultures, and traditions of people can be explicitly seen. Figure 2 shows a knowledge graph analysis of 50 traditions common culture.

As can be seen in Figure 2, the knowledge graph shows the relations of 50 traditions in the three studied countries (i.e., Thailand, Laos and Cambodia). These traditions are only part of all the traditions in the GMS. In addition, it can be seen, as presented by the knowledge graph, that the names of the traditions were different, although they referred to the same traditions. For example, these names (i.e., Boun Ork Phansa, End of Buddhist Lent, Boun Ork Phansa, Bonn Chenh Vassa, etc.) refer to the same tradition, which marks the last day of the observance of Vassa around October. The numbers and black bars show the relative density values, which can increase the accuracy of the search. Thus, this research adopted ontology concepts to develop knowledge structures so that the search system

could search through concepts, properties, and relationships to overcome the semantic gap in knowledge levels.

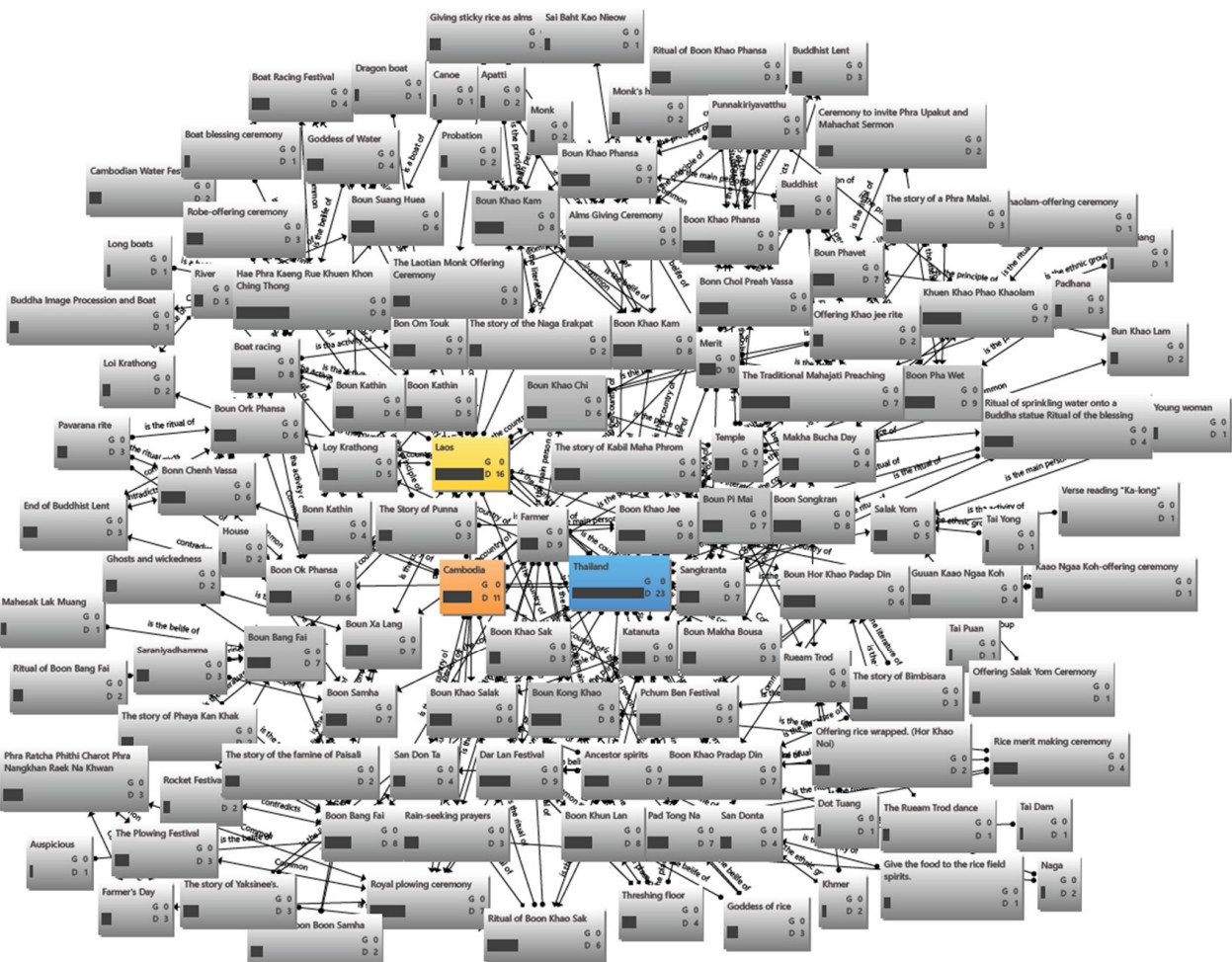

**Figure 2.** The systematic mapping of knowledge of the common culture of countries in the GMS.

*2.2. Ontology Development*

The ontology development was based on the concept of Noy and McGuinness [15], and Hozo Ontology Editor [27] was employed. This process comprised of the following steps.

1.  Determine the scope;
2.  Consider reuse;
3.  Enumerate terms;
4.  Define classes;
5.  Define properties;
6.  Define constraints;
7.  Create instances.

*2.3. Ontology Evaluation*

The ontology evaluation was to confirm the academic validity of the relevant content of the topic on the common culture of the GMS countries and the validity of structural characteristics and ontology description. This study used the assessment by humans against a set of criteria. The evaluation form was developed based on Gomez–Perez's concept of ontology evaluation [29,30], and the structure of the ontology was evaluated according to the purpose or scope of the formation of ontology. This was evaluated by experts on the common culture of the GMS countries and the validity of the structural characteristics

and ontology description. A purposive sampling method was employed to select three evaluators with the following qualifications: being knowledgeable and experienced in conducting studies on ontology and semantic web as well as having published academic papers in national and internationally recognized journals from the last five years.

To evaluate the quality of ontology, descriptive statistics were used, along with a five-point Likert scale [31] as a criterion to interpret the quality level of the developed ontology. Additionally, user recommendations were drawn on to improve the ontology; the quality evaluation results of the development of the ontology were interpreted by the criteria for the mean score evaluation [32]. Moreover, we used the application-based ontology evaluation to conduct an ontology assessment and measure the effectiveness of information retrieval with a semantic search prototype.

### 3. Results

The results of the study on the development of an ontology for the knowledge of the common culture of the GMS countries are presented according to each process of ontology development.

### 3.1. Determine the Scope

The ontology, which was developed in this study, covered the traditions common culture in the GMS countries. The scope of the research was limited to traditions within 12 months of Thailand, the Lao People's Democratic Republic, and the Kingdom of Cambodia. The common culture was presented in the form of concepts and terms. Therefore, knowledge related to the common culture of the GMS countries was described by providing the definitions, meaning, and defining the properties and attributes, sample data, and related relationships between the classes and terms. Moreover, the structure of ontology was determined by defining the main domain; existing knowledge of the traditions common culture was used to define the classes.

The purpose of ontology development was to provide primary data in the development of a search engine—a semantic search system. The system must be able to look for information from cultural commonalities according to the concept of four characteristics of common culture: History; Values and Principles; Purpose and Sense of mission; and Symbols and Boundaries [6].

### 3.2. Consider Reuse

This research aimed to develop a prototypical search engine for the common culture of the GMS countries. In particular, a search engine was developed as a semantic web by using ontology. This research used methods for restructuring and adding relevant knowledge. A domain ontology is a collection of keywords that can be used to describe domain concepts and to describe data collection patterns as well as indices that can be shared between applications [33]. Some parts of the existing ontologies related to ontology development in this study were adapted for the scope of this study, as detailed below.

1. The ontology for the knowledge of the Twelve-month Isan Merit-Making Traditions, Heed Sip Song, [18,19] is the development of a specialized ontology, the contents of which focused only on the traditions in northeastern Thailand. It consisted of ten main classes related to knowledge of the traditions: Custom, Language, Place, Region, Person, Time, Ritual, Objective, Activity, and Belief. When comparing the ontology of The Twelve-month Isan Merit-Making Traditions with that of this study with different scopes, it was found that there were seven similar classes including Activity, Belief, Objective/Purpose, Person, Place, Ritual, and Time.

2. The ontology for knowledge about ethnic groups [24,34–36] explained the overall concept of ethnic groups and had a different scope compared to the study. Its contents were related to religions, beliefs, accommodations, costumes, social organizations, art, and the history of a particular ethnic group. Therefore, to apply the ontology of ethnic groups, only qualities of the ethnic group class were applied, namely ethnonym,

exonym, and autonym, which were the subclass of ethnic groups under common ecological characteristics.

3. The Event Ontology was developed by the Center for Digital Music, University of London [20]; specifically, the classes and class descriptions related to time and place were applied. The existing ontology may not be completely applicable due to differences in the scopes and contexts of the domain in the study. Therefore, only some applicable classes were drawn on and partly improved not to affect the main concept and effectiveness, and its accuracy was validated by experts in the ontology evaluation process.

4. The Location Ontology was developed by the DBpedia Team. the location class was implemented, consisting of the country class (LocationCountry) and the city class (LocationCity) [21].

### 3.3. Enumerate Terms

The researcher synthesized a list of terms in line with the related documents and then analyzed their meanings to reduce duplication among the three languages. The synthesis and analysis were mainly based on the definitions of the terms, which were translated into English, the language used to develop the system. Subsequently, the essential words in ontology were distributed into three features: (1) Terms; (2) Properties; and (3) Key Definitions of Ontology and Defining Term Properties [15,37].

### 3.4. Define Classes

In defining the classes related to the study, there was a total of 52 classes including 15 main classes: common culture, history, belief, purpose, location, ritual, activity, literature, values, place, time, principle, person, equipment, and ethnic group. The traditions common culture was related to all other classes presented in the Semantic Web Table Specification to comply with W3C's RDF standards [38]. The ontology data were presented according to the description example, as shown in Tables 1 and 2.

**Table 1.** The ontology of the traditions common culture in the GMS countries.

| Term Name | | Definition |
|---|---|---|
| **Class** | **Subclass** | |
| Common culture | | Common culture |
| | Ethnic common culture | Ethnic common culture |
| | Belief common culture | Belief common culture |
| | Historical common culture | Historical common culture |
| | Language common culture | Language common culture |
| | Religious common culture | Religious common culture |
| | Architectural common culture | Architectural common culture |
| | Traditions common culture | Traditions common culture |
| Location | | The location of the tradition |
| | Location country | Country the tradition is located. |
| | Location city | City the tradition is located |
| History | | History of traditions |
| | Cause of Religious Beliefs | History of traditions that come from religious beliefs. |
| | Cause of Superstition | History of traditions that come from Superstition beliefs. |
| | Cause of Geographic Environment | History of traditions that come from geographic environment. |
| Ethnic Group | | Ethnic groups related to traditions |
| Belief | | Beliefs related to traditions |
| | Buddhism | Buddhism beliefs related to traditions |
| | Superstition | superstition related to traditions |

**Table 1.** *Cont.*

| Term Name | | Definition |
|---|---|---|
| **Class** | **Subclass** | |
| Literatures | | Literature related to traditions |
| | Oral Literature | Oral literature related to traditions |
| | Written Literature | Written literature related to traditions |
| Values | | Values reflected in traditions |
| | Forgiveness | Forgiveness reflected in traditions |
| | Gratitude | Gratitude reflected in traditions |
| | Unity | Unity reflected in traditions |
| | Perseverance | Perseverance reflected in traditions |
| | Sacrifice | Sacrifice reflected in traditions |
| Principle | | Principles related to traditions |
| | People Principle | Layperson's principle related to traditions |
| | Priest Principle | Buddhist priest's principles related to traditions |
| Purpose | | Purpose of traditions |
| | Livelihood Purpose | Livelihood purpose |
| | Morale For Living Purpose | Morale for living purposes |
| | Gratitude Purpose | Gratitude purpose |
| | Buddhism Purpose | Buddhism purpose |
| Place | | Place of the establishment of traditions |
| | Public Space | Public space |
| | Domestic Space | Domestic space |
| Equipment | | Important equipment used in traditions |
| | Foods | Foods used in traditions |
| | Dharma Offerings | Dharma offerings used in traditions |
| | Decoration | Decorations used in traditions |
| | Tools | Tools used in traditions |
| | Monk's Utensils | Monk's utensils used in traditions |
| Time | | Time of establishment of traditions |
| | Calendar | Solar calendar |
| | Lunar Time | Lunar time |
| Activity | | Activity performed by humans during traditions |
| | Belief Activities | Traditions activities related to belief |
| | Religious Activities | Religious activities in traditions |
| | Carnival Activities | Carnival activities in traditions |
| | Family Activities | Tradition activities in a family |
| Ritual | | Rituals related to traditions |
| | Traditional Ritual | Traditional rituals |
| | Religious Ritual | Religious rituals |
| Person | | Person related to the establishment of traditions |
| | Officiant | Person as a leader to perform rituals |
| | Participants | Person who attends rituals |

**Table 2.** An example of creating instances.

| Role Concept | Class Constraint | Instance |
|---|---|---|
| Tradition id | integer | 1 |
| Tradition name | string | Boon Khun Lan |
| Other name | string | Boon Kum Khao Yai |
| Has Ethnic group | string | |
| Has Country | string | Thailand |

**Table 2.** *Cont.*

| Role Concept | Class Constraint | Instance |
|---|---|---|
| Has History | string | The origin of the merit-making ceremony in Thailand is derived from activities after the end of the harvest season. After completing the rice harvesting, farmers gather the rice to form a "pile of rice" in their fields. If the pile is high, that means they have high productivity and fertile paddy fields. The owner is delighted and joyful, so they make merit with the hope that they will have higher productivity in the following year. That is called "Koon", or make it bigger and higher"; the word "Koon" comes from "Kham Koon", which means to contribute for the better and help them grow. |
| Has Purpose | string | To celebrate the rice harvest season and show appreciation to the spirits for their help with the cultivation. Making charity will lead to higher productivity in the following year. |
| Has Place | string | Rice field |
| Has Literatures | string | The legend of Phi Ta Hak |
| Has Equipment | string | Rice pile |
| Has Belief | string | Guardian Spirit Rice Fields |
| Has Times | string | January |
| Has Activity | string | To offer food to the monk |
| Has Person | string | Farmer |
| Has Principle | string | Katannuta |
| Has Ritual | string | The Big Rice Heap Merit-Making Ritual |
| Has Values | string | Gratitude |

*3.5. Define Properties and Define Constraints*

This step involves defining the conditions or criteria for the validation of class properties. The classes of the traditions common culture had a relationship with all of the other classes in the ontology, which helped link the data in the semantic search of the traditions common culture of the GMS countries. Based on the definition of the properties of classes or relationships between concepts, it was found that 73 concepts had an is-a property, while 15 concepts had a part-of property, and 52 had attribute-of property, as displayed in Figure 3.

*3.6. Create Instances*

This step concerns the creation of instances of concepts or terms defined in the ontology, instances of classes, and subclasses. Through instances, relationships can be designed among all of the attributes of the respective ontology. Table 2 shows the examples of instances of Boon Khun Lan. All instances were configured for all classes and subclasses. Each class had many memberships that act as instances. For example, the Belief class included Merit, Goddess of Rice, Mahesak Lak Muang, Naga, Ghosts, Ancestor spirits, etc.

*3.7. Ontology Evaluation*

Ontology evaluation was conducted to verify the structure and descriptions of the ontology. The structure of the ontology was evaluated by three ontology experts (see Table 3). Prototyping and hypotheses testing of the application-based ontology evaluation was also conducted (see Tables 4 and 5).

As shown in Table 3, the overall evaluation results of Determine the Scope were at a high level ($\bar{x}$ = 4.22). Similarly, those of Define Classes/Concepts were at a high level ($\bar{x}$ = 3.83). When each item was taken into account separately, it was found that the

evaluation results of defining concepts, classifying the superclass, classifying the sub-class, and defining the datatype were at a high level while those of defining terms were at a moderate level. For Define Properties, the evaluation results of each aspect including defining related properties, defining relationships between related concepts, and having consistent relationships were at a high level; thus, the overall evaluation results were at a high level ($\bar{x}$ = 4.11). The overall evaluation results of Creating Instances (e.g., defining instances in terms of definitions and terms and grammar) were at a high level ($\bar{x}$ = 4.00). Finally, the overall evaluation results of Application to Ontology Development were at a high level ($\bar{x}$ = 4.17).

In this research, we used the application-based ontology evaluation to conduct an ontology assessment and measure the effectiveness of information retrievals with a semantic search prototype (see Figure 4) In this prototype, we used the Ontology-based Application Management Framework (OAM). This application provides reusable and configurable application templates and can build prototypes. Moreover, the Application template is capable of rapid prototyping and hypotheses testing. The framework allows Web API to support a more advanced application development [10]. The ontology can be evaluated using information retrieval such as precision, recall, and the F-measure. We evaluated the system performance by calculating: (1) the precision value to determine the fraction of relevant retrieved document; (2) the recall value to determine whether recall is the fraction of the relevant documents that are retrieved; and (3) the F-measure using an equation. The performances of the keyword search of the semantic search system with the precision, recall, and F-measure were 0.90, 0.88, and 0.85, respectively (see Table 4); the evaluation results with the advanced search were 1.00, 1.00, and 1.00, respectively (see Table 5).

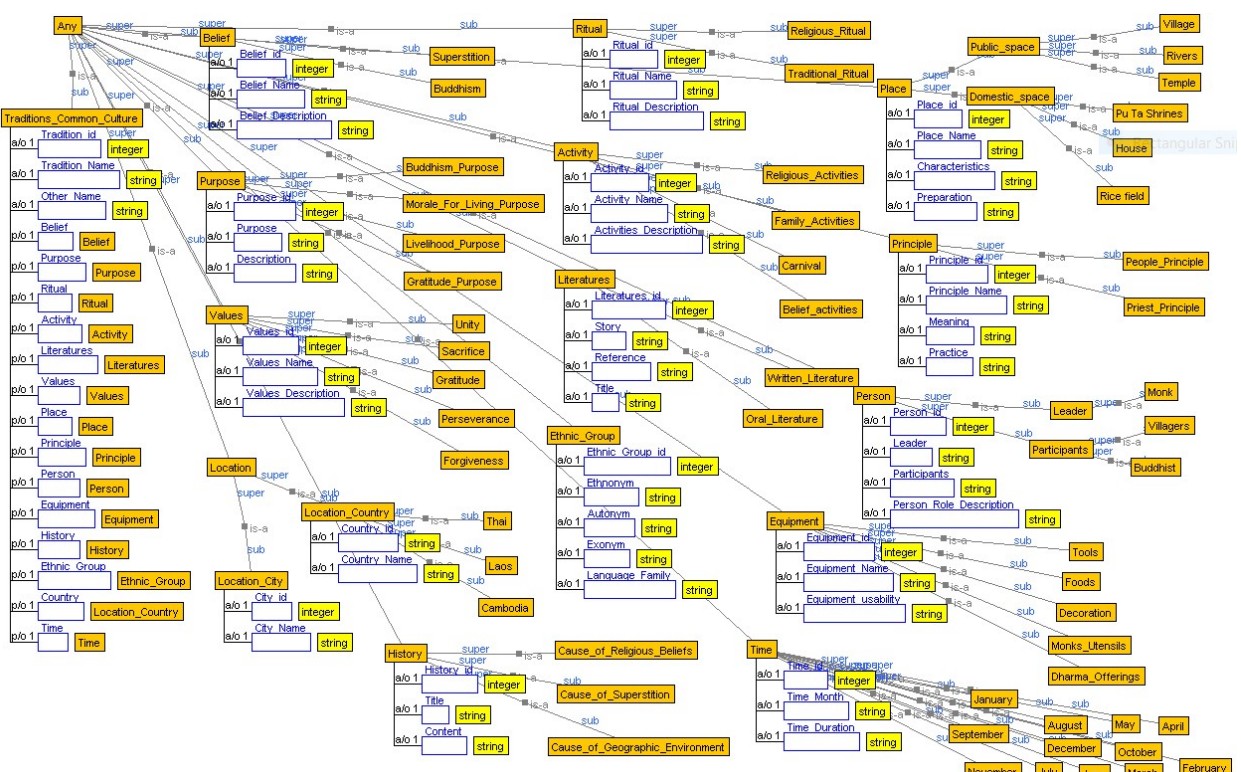

**Figure 3.** Defining the class properties of the traditions common culture.

**Table 3.** The evaluation of traditions common culture of countries in the Greater Mekong Subregion ontology results by experts.

| No. | Statements | Levels of Agreement (*N* = 3) | |
|---|---|---|---|
| | | $\bar{x}$ | Levels |
| | *Determine the Scope* | | |
| 1 | Ontology matches the scope of knowledge determined in this study. | 4.33 | High |
| 2 | Ontology is suitable and covers the scope of knowledge in this study. | 4.33 | High |
| 3 | Ontology can be applied to the development of ontology of the traditions common culture of the GMS countries. | 4.00 | High |
| | Total | 4.22 | High |
| | *Define Classes/Concepts* | | |
| 4 | Ontology defines concepts that can describe knowledge properly. | 4.33 | High |
| 5 | Ontology classifies the superclass appropriately. | 4.33 | High |
| 6 | Ontology classifies the sub-class appropriately. | 4.00 | High |
| 7 | Ontology defines the datatype appropriately. | 3.67 | High |
| 8 | Ontology defines terms appropriately. | 3.33 | Moderate |
| 9 | Ontology defines class constraints appropriately. | 3.33 | Moderate |
| | Total | 3.83 | High |
| | *Define Properties* | | |
| 10 | Ontology defines the related properties to describe concepts appropriately. | 4.00 | High |
| 11 | Ontology defines the relationships between related concepts to describe concepts appropriately. | 4.33 | High |
| 12 | Ontology has consistent relationships. | 4.00 | High |
| | Total | 4.11 | High |
| | *Create Instances* | | |
| 15 | Ontology properly defines instances with common definitions. | 4.00 | High |
| 16 | Ontology properly defines instances with the correct terms and grammar. | 4.00 | High |
| | Total | 4.00 | High |
| | *Application to Ontology Development* | | |
| 17 | Ontology is accurate and reliable. | 4.33 | High |
| 18 | Ontology can be reused to develop other systems. | 4.00 | High |
| | Total | 4.17 | High |

**Table 4.** The performance of the semantic information retrieval system with keyword search.

| Queries | Retrieved | Relevant Retrieved | Relevant in the Collection | Precision | Recall | F-Measure |
|---|---|---|---|---|---|---|
| Rocket Festival | 2 | 2 | 2 | 1.00 | 1.00 | 1.00 |
| Buddhist Lent | 8 | 3 | 3 | 0.38 | 1.00 | 0.55 |
| End of Buddhist Lent | 4 | 3 | 3 | 0.75 | 1.00 | 0.86 |
| Songkran | 5 | 5 | 5 | 1.00 | 1.00 | 1.00 |
| Alms Giving | 2 | 2 | 2 | 1.00 | 1.00 | 1.00 |
| Makha Bucha | 2 | 2 | 2 | 1.00 | 1.00 | 1.00 |
| Mahachat | 1 | 1 | 3 | 1.00 | 0.33 | 0.50 |
| Rain | 9 | 8 | 8 | 0.89 | 1.00 | 0.94 |
| Boon Samha | 1 | 1 | 2 | 1.00 | 0.50 | 0.67 |
| Kathin | 3 | 3 | 3 | 1.00 | 1.00 | 1.00 |
| **Average** | | | | 0.90 | 0.88 | 0.85 |

**Table 5.** The performance of the semantic information retrieval system with advanced search.

| Condition | | Retrieved | Relevant Retrieved | Relevant in the Collection | Precision | Recall | F-Measure |
|---|---|---|---|---|---|---|---|
| Class | Sub-Class | | | | | | |
| Equipment | Foods | 8 | 8 | 8 | 1.00 | 1.00 | 1.00 |
| Activity | Religious_ Activities | 8 | 8 | 8 | 1.00 | 1.00 | 1.00 |
| Belief | Goddess_of_rice | 3 | 3 | 3 | 1.00 | 1.00 | 1.00 |
| Ethnic_Group | Khmer | 2 | 2 | 2 | 1.00 | 1.00 | 1.00 |
| History | Cause_of_ Religious_Beliefs | 22 | 22 | 22 | 1.00 | 1.00 | 1.00 |
| Literatures | Annals | 1 | 1 | 1 | 1.00 | 1.00 | 1.00 |
| Country | Thai | 23 | 23 | 23 | 1.00 | 1.00 | 1.00 |
| Person | Farmer | 16 | 16 | 16 | 1.00 | 1.00 | 1.00 |
| Place | Rivers | 5 | 5 | 5 | 1.00 | 1.00 | 1.00 |
| Principle | Priest Principle | 2 | 2 | 2 | 1.00 | 1.00 | 1.00 |
| Purpose | Gratitude_ Purpose | 8 | 8 | 8 | 1.00 | 1.00 | 1.00 |
| Ritual | Religious Ritual | 14 | 14 | 14 | 1.00 | 1.00 | 1.00 |
| Time | February | 5 | 5 | 5 | 1.00 | 1.00 | 1.00 |
| Values | Unity | 6 | 6 | 6 | 1.00 | 1.00 | 1.00 |
| **Average** | | | | | 1.00 | 1.00 | 1.00 |

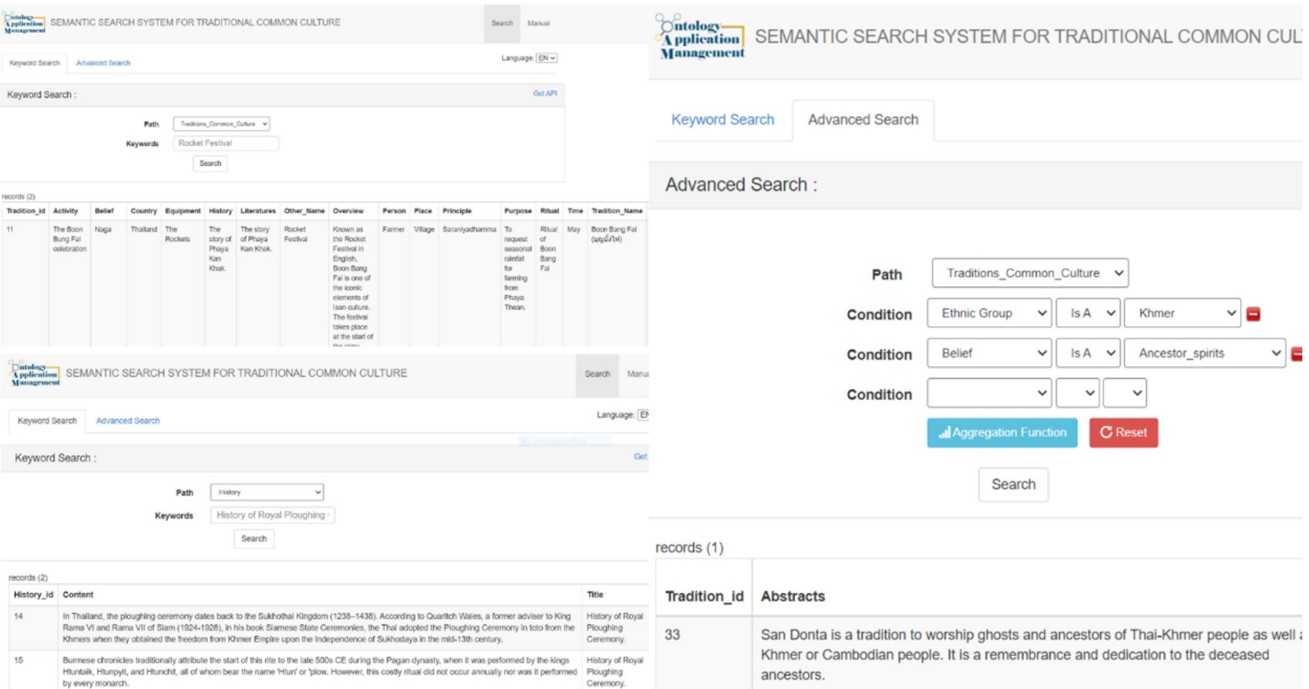

**Figure 4.** The prototype of the semantic search system using ontology.

As can be seen in Table 5, the results of the advanced search performance evaluation showed a mean F-measure of 1.00. This indicates that the semantic search system developed from this ontology knowledge base has high browsing efficiency.

## 4. Discussion

The study and development of the semantic search on the traditions common culture of the GMS countries was carried out in compliance with the policy of Managing the Intangible Cultural Heritage [39,40]. As stipulated in the policy, it is necessary to develop a list of intangible cultural heritage, promote research, and collect related documents or evidence for the development of safeguarding measures and for education. In addition, the knowledge organization in this study was in sync with Thailand's Management of the Intangible Cultural Heritage [40]. That is, a proposal was made to build a database of Thailand's cultural heritage with accurate and quality data verified by experts. It was also proposed to collect and organize the data systematically to create a knowledge base for education and research purposes; these data should be linked and constantly updated. The dissemination of knowledge and a knowledge base must be effective and comprehensive while a source of information should be accessed with ease via a knowledge base or a source of knowledge. These processes are regarded as approaches to the management of intangible cultural heritage [41–43]. The study of the traditions common culture of the GMS countries was compliant with the aforementioned policy; that is, the traditions common culture of the GMS countries, which is a form of intangible cultural heritage, was systematically organized and presented.

## 5. Conclusions

Based on the results of the study, the knowledge structure on the traditions common culture of the GMS countries can be drawn on to develop research trends on more specific issues within this topic. In addition, it could be used to determine headings or keywords for the classification of related information resources and used as keywords to facilitate information searches in other databases. The ontology developed in this study would also serve as the foundation for the development of the semantic search of knowledge about the traditions common culture of countries in the GMS. Specifically, it could be used as a model search system for cultural information, which facilitates information search in a specific field. Finally, researchers can develop similar search systems on other issues, which would serve as a source for storing and retrieving knowledge.

**Author Contributions:** Conceptualization, S.H. and K.K.; Methodology, S.H. and K.K.; Validation, S.H.; Investigation, S.H.; Writing—original draft preparation, S.H.; Writing—review and editing, S.H. and K.K.; Supervision, S.H.; Project administration, S.H. All authors have read and agreed to the published version of the manuscript.

**Funding:** This research was supported by the Graduate School Khon Kaen University, Thailand. Grant number 621S217.

**Institutional Review Board Statement:** Not applicable.

**Informed Consent Statement:** Not applicable.

**Data Availability Statement:** Not applicable.

**Conflicts of Interest:** The authors declare no conflict of interest. The funders had no role in the design of the study; in the collection, analyses, or interpretation of data; in the writing of the manuscript; or in the decision to publish the results.

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
