# Peer review of "Development of Ontology for Knowledge of Traditions Common Culture of Countries in the Greater Mekong Subregion"

_informatics, doi:10.3390/informatics9030058_

Round 1

Reviewer 1 Report

The authors have made a reasonable attempt to address my comments. There is scope for improvement but I will let that be.

Author Response

Thank you very much for your constructive suggestions for editing the article. These great reviews are very helpful in enhancing the article and polishing the writing.

Therefore, based on the information we have received, Reviewer 1 has accepted the revision of our writing.

We would like to respond to the Reviewer 2. We have tried to modify the manuscript based on your feedback. We thank you very much for your professional work and time.

Response to Reviewer 1 Comments

Point 1: The authors have made a reasonable attempt to address my comments. There is scope for improvement but I will let that be.

Response 1: Thank you for your decision.

Reviewer 2 Report

Article defines an ontology adequately. It is missing a reference to previous or partial ontologies on which to compare the current work and specify its improvement with respect to them.

For this reason, I recommend an extension of the introduction or a paragraph that specifies the non-existence of previous ontologies.

Author Response

Thank you very much for your constructive suggestions for editing the article. These great reviews are very helpful in enhancing the article and polishing the writing.

Therefore, based on the information we have received, Reviewer 1 has accepted the revision of our writing.

We would like to respond to the Reviewer 2. We have tried to modify the manuscript based on your feedback. We thank you very much for your professional work and time.

Response to Reviewer 2 Comments

 Point 1: Article defines an ontology adequately. It is missing a reference to previous or partial ontologies on which to compare the current work and specify its improvement with respect to them. For this reason, I recommend an extension of the introduction or a paragraph that specifies the non-existence of previous ontologies.

Response 1: Thank you for your suggestion. We added already sentences following your suggestion into the introduction section at Paragraph 7 and 8, line 113 - 160. 

This manuscript is a resubmission of an earlier submission. The following is a list of the peer review reports and author responses from that submission.

Round 1

Reviewer 1 Report

The paper seeks to “…create an ontology that represents the knowledge of the common culture of countries in the Greater Mekong Subregion and to gain a perspicuous understanding of the knowledge about traditions common culture of those countries in this region.” It applies a seven-step method using the Hozo Ontology Editor.

The paper addresses an important problem. However, its mechanical application of the method leads to results that are not meaningful, at least to this reviewer. I cannot interpret the meaning of the last three sentences of the abstract and their elaboration in the paper. Part of the problem may be due to deficiencies in the language, but another part of it appears to be the mechanical approach depending on the method and the tool. It does not appear to be informed by a theoretical or conceptual framework, or the desire to develop one based on the grounded approach.

I tried to make sense of the results, especially Table 1 and the following figures. (The figures in the manuscript are very difficult to read.) The Figure 1 below is a reorganization of the Term Names in Table 1 of the paper. (That is how I represent ana ontology.) It reclassifies the terms into dimensions (columns) and taxonomies of the dimensions. It is better formed than Table 1 and avoids errors of logical typing (confusion of a class with its subclass). It also helps enumerate the components of knowledge of common culture systematically. Three components are illustrated in the figure. Each of these components may be instantiated. The ontology encapsulates 1225 components. The instantiations of these would encapsulate the knowledge of common culture. Some of the components may not be instantiated, and some may have multiple instantiations.

Systematically mapping “…the knowledge of the common culture of countries in the Greater Mekong Subregion…” may help “…to gain a perspicuous understanding of the knowledge about traditions common culture of those countries in this region.” Studying the knowledge through the lens in Figure 1 would highlight the emphases and gaps in the knowledge, and similarities and differences between the subregions. Presumably, that is the object of the presented research.

Author Response

Thank you very much for the constructive suggestions of the paper revision. These great reviews are quite helpful to make the paper better and polish the writing. We appreciate a lot for your professional work and time. Also, We respond to the comments one by one as follows. Please see the attachment.

Best regards,
Kanyarat Kwiecien

Reviewer 2 Report

First of all congratulations to the authors for the article. The creation of an ontology on such a large collective is always problematic. Normally, the problem usually lies in its complexity and the difficulty of finding a test case on which to instantiate it. In the article I find no reference to the test method beyond a personal validation by three experts that I do not consider a reliable method to validate such an extensive ontology. I recommend finding a group of documents on which to apply the ontology to be able to validate it correctly on a set of users. This method is usually the best way to validate large ontologies.

Author Response

(The authors gave the same response as above.)

Reviewer 3 Report

The paper aims to create an ontology that represents the knowledge of the common culture of countries in the Greater Mekong Subregion and to gain a perspicuous understanding of the knowledge about traditions and common culture of those countries in this region.

In my opinion, something needs to be improved before the publication. 

A discussion on knowledge organization is currently presented in the introduction section. To improve the paper, a distinct section on comparison among the related works on ontologies for cultural heritage needs to be added. To improve the paper, the authors should provide a discussion characterizing each work, its benefits and drawbacks, and a comparison among the related works to justify the necessity of the study proposed in the paper. Please state clearly and precisely in the paper what makes this work original.

To improve the readability of the paper, an outline needs to be added at the end of the introduction section.

Author Response

Thank you very much for the constructive suggestions of the paper revision. These great reviews are quite helpful to make the paper better and polish the writing. We appreciate a lot for your professional work and time. Also, We respond to the comments one by one as follows. Please see the attachment.

Round 2

Reviewer 1 Report

The authors have added a few sentences here and there in response to the request for major revision by all the three reviewers, including me. The response is perfunctory, cursory, and inadequate. My reaction during the first review was that the paper is a mechanical application of a method without meaningful context, explanation, and interpretation. I wanted to give the authors an opportunity to revise and make it more meaningful. The authors have chosen to make only marginal modifications in response to my and the other reviewers comments. Perhaps they have not understood the import of the comments, or they do not agree with them. In the present form the paper must be rejected.

Reviewer 3 Report

In the revised paper, the authors have solved the changes I proposed.